# On Exact Solutions of the Inner Optimization Problem of Adversarial Robustness

## Abstract

In this work, we propose a robust framework that employs adversarially robust training to safeguard the ML models against perturbed testing data. Our contributions can be seen from both computational and statistical perspectives. Firstly, from a *computational/optimization* point of view, we derive the ready-to-use exact solution for several widely used loss functions with a variety of norm constraints on adversarial perturbation for various supervised and unsupervised ML problems, including regression, classification, two-layer neural networks, graphical models, and matrix completion. The solutions are either in closed-form, or an easily tractable optimization problem such as 1-D convex optimization, semidefinite programming, difference of convex programming or a sorting-based algorithm. Secondly, from *statistical/generalization* viewpoint, using some of these results, we derive novel bounds of the adversarial Rademacher complexity for various problems, which entails new generalization bounds. Thirdly, ~~we validate our approach by showing significant performance improvement~~we perform some sanity-check experiments on real-world datasets ~~over various gradient ascent based baselines~~ for supervised problems such as regression and classification, as well as for unsupervised problems such as matrix completion and learning graphical models, with very little computational overhead.

## 1 Introduction

Machine learning models are used in a wide variety of applications, such as image classification, speech recognition, and self-driving vehicles. The models employed in these applications can achieve a very high training time accuracy but can fail spectacularly in making trustworthy predictions on test data. Thus, it becomes important for existing machine learning models to be adversarially robust to avoid poor performance and have better generalization on test data. Our contribution in this work encompasses both the *computational/optimization* and *statistical/generalization* aspects of adversarial training for various supervised and unsupervised learning problems.

Firstly, from a *computational/optimization* perspective, we focus on deriving *the exact optimal solution* of the inner maximization optimization arising in adversarial training. In particular, we provide ready-to-use results for a wide variety of loss functions and various norm constraints (See Table 1). Moreover, unlike other domain specific works such as (Jia & Liang, 2017; Li et al., 2016; Belinkov & Bisk, 2018; Ribeiro et al., 2018) in natural language processing and (Hendrycks et al., 2021; Alzantot et al., 2018) in computer vision, we aim to provide an adversarially robust training model which covers a large class of machine learning problems.

From a *statistical/generalization* perspective, our contributions include providing upper and lower bounds for adversarial Rademacher complexity ($\mathcal{O}(1/\sqrt{n})$). These bounds are based on the results summarized in Table 1. By analyzing the adversarial Rademacher complexity, we can infer the generalization aspect discussed in the introduction of Section 5.1. In this regard, we propose novel adversarial Rademacher complexity bounds for various ML problems like linear regression (applicable for any general norm), matrix completion, and max-margin matrix completion. Additionally, we propose bounds in Theorem 14 for linear classifiers, which can be seen as a generalization of the $\ell_\infty$ norm (Yin et al., 2019) or any specific $p$-norm (Awasthi et al., 2020).

Table 1: A summary of our results for various loss functions and norm constraints which are used in a wide variety of applications.

| | Problem | Loss function | Norm constraint | Prior results | Our solution |
|---|---|---|---|---|---|
| Warm up | Regression | Squared loss | Any norm | Euclidean norm (Xu et al., 2008) | Closed form, Theorem 3 |
| | Classification | Logistic loss | Any norm | Euclidean norm (Liu et al., 2020) | Closed form, Theorem 4 |
| | Classification | Hinge loss | Any norm | None | Closed form, Theorem 21 |
| Main results | Classification | Two-Layer NN with convex and nonconvex activations | Any norm | ReLU activation (Awasthi et al., 2024) | Difference of convex functions, Theorem 5 |
| | Graphical Models | Log-likelihood | Euclidean | None | 1-D optimization, Theorem 6 |
| | Graphical Models | Log-likelihood | Entry-wise $\ell_\infty$ | None | Semidefinite programming, Theorem 7 |
| | Matrix Completion | Squared loss | Frobenius | None | Closed form, Theorem 8 |
| | Matrix Completion | Squared loss | Entry-wise $\ell_\infty$ | None | Closed form, Corollary 9 |
| | Max-Margin MC | Hinge loss | Frobenius | None | Sorting based algorithm, Theorem 10 |
| | Max-Margin MC | Hinge loss | Entry-wise $\ell_\infty$ | None | Closed form, Corollary 11 |

To make the above-discussed computational and statistical contributions, we propose closed-form solutions or an easily tractable optimization problem for computing the adversarial perturbation in these multiple ML problems. These solutions summarized in Table 1 are in closed form or can be obtained from a one-dimensional dual problem, semidefinite programming (SDP), or a sorting-based algorithm which are quite unexpected and novel in the context of adversarial perturbation. The solution to the inner maximization for various problems summarized in Table 1 allows to get better robustness in a computationally cheap manner.

~~In this work, we view adversarially robust training in a practical way and reduce the run time from $\mathcal{O}(nTL)$ to $\mathcal{O}(nT)$, whe~~

~~For problems with a complex objective function with no closed-form solution, our approach of having computationally tract~~

**Our Contributions.** Broadly, we make the following contributions through this work:

- **Adversarially robust formulation:** We use the adversarially robust training framework of Yin et al. (2019) using worst case adversarial attacks. Under this framework, we analyze several supervised and unsupervised ML problems, including regression, classification, two-layer neural networks, graphical models, and matrix completion. The solutions are either in closed-form, 1-D optimization, semidefinite programming, difference of convex programming, or a sorting-based algorithm.
- **Computational/Optimization front:** We provide a plug-and-play solution which can be easily integrated with existing training algorithms. This is a boon for practitioners who can incorporate our method in their existing models with minimal changes. As a conscious design choice, we provide computationally cheap solutions for our optimization problems.
- **Statistical/Generalization front:** On the theoretical front, we provide a systematic analysis for several loss functions and norm constraints, which are commonly used in applications across various domains. Table 1 provides a summary of our findings in a concise manner. Using some of these results, we further provide novel lower and upper bounds of the adversarial Rademacher complexity ($\mathcal{O}\left(1/\sqrt{n}\right)$) for various problems, which entails novel generalization bounds.
- **Real world experiments:** We further ~~validate our results by conducting~~ perform some sanity-check experiments on several real-world datasets. We show that our plug-and-play solution performs better (most of the time, in terms of test metrics and/or runtime) as compared to the fast gradient sign method (FGSM) (Goodfellow et al., 2015), projected gradient descent (PGD), and TRADES (Zhang et al., 2019).

## 2 Preliminaries

For any general prediction problem in machine learning (ML), consider we have $n$ samples of $(\mathbf{x}, y)$, where we try to predict $\mathbf{y} \in \mathcal{Y}$ from $\mathbf{x} \in \mathcal{X}$ using the function $f : \mathcal{X} \to \mathcal{Y}$. Assuming that the function $f$ can be parameterized by some parameter $\mathbf{w}$, we minimize a loss function $l(\mathbf{x}, y, \mathbf{w})$ to obtain an estimate of $\mathbf{w}$ from

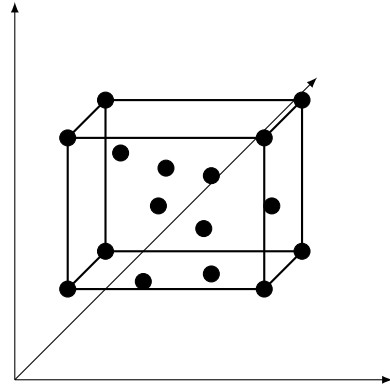

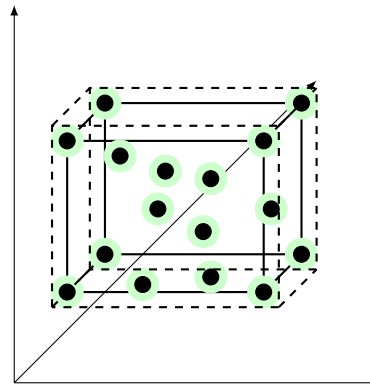

Figure 1: Training domain    Figure 2: Worst case adversarial attack domain

Figure 1 shows the domain for clean training points while the dashed cube in Figure 2 shows the worst case adversarial attack domain (slightly bigger than the original training domain). Each new worst case adversarially attacked point is judiciously picked from within the green spheres around the corresponding clean training point with radius $\epsilon$ in a predefined norm.

$n$ samples:

$$\hat{\mathbf{w}} = \arg\min_{w} \frac{1}{n} \sum_{i=1}^{n} l(\mathbf{x}^{(i)}, y^{(i)}, \mathbf{w}) \tag{1}$$

where $(\mathbf{x}^{(i)}, y^{(i)})$ represents the $i^{\text{th}}$ sample.

Intuitively, with no prior information on the shift of the testing distribution, it makes sense to be prepared for absolutely worst-case scenarios. We incorporate this insight formally in our proposed *adversarially robust training model*. At each iteration of the training algorithm, we generate worst-case adversarial samples using the current model parameters and "clean" training data within the bounds of a maximum norm. The model parameters are updated using these worst-case adversarial samples, and the next iteration is performed. Figure 1 and Figure 2 provide a geometric interpretation of our training process.

Before proceeding to the main discussion, we briefly discuss the notations and basic mathematical definitions used in the paper.

**Notation:**  We use a lowercase alphabet such as $x$ to denote a scalar, a lowercase bold alphabet such as $\mathbf{x}$ to denote a vector and an uppercase bold alphabet such as $\mathbf{X}$ to denote a matrix. The $i^{\text{th}}$ entry of the vector $\mathbf{x}$ is denoted by $\mathbf{x}_i$. The superscript star on a vector or matrix such as $\mathbf{x}^\star$ denotes it is the optimal solution for some optimization problem. A general norm for a vector is denoted by $\|\mathbf{x}\|$, and its dual norm is indicated by a subscript asterisk, such as $\|\mathbf{x}\|_*$. The set $\{1, 2, \ldots, n\}$ is denoted by $[n]$. A set is represented by capital calligraphic alphabet such as $\mathcal{P}$, and its cardinality is represented by $|\mathcal{P}|$. For a scalar $x$, $|x|$ represents its absolute value.

**Definition 1.** The dual norm of a vector, $\|\cdot\|_*$ is defined as:

$$\|\mathbf{z}\|_* = \sup\{\mathbf{z}^\mathsf{T}\mathbf{x} \mid \|\mathbf{x}\| \leq 1\} \tag{2}$$

**Definition 2.** Let $\|\cdot\|_*$ is the dual norm to $\|.\|$. The sub-differential of a norm is defined as:

$$\partial \|\mathbf{x}\| = \{\mathbf{v} : \mathbf{v}^\mathsf{T}\mathbf{x} = \|\mathbf{x}\|, \|\mathbf{v}\|_* \leq 1\} \tag{3}$$

In this work, we propose *plug-and-play* solutions for various ML problems to enable adversarially robust training. By plug-and-play solution, we mean that any addition to the existing algorithm comes in terms of a closed-form equation or as a solution to an easy-to-solve optimization problem. Such a solution can be integrated with the existing algorithm with very minimal changes.

# 3   Warm Up

In this section, we formally discuss our proposed approach of adversarially robust training on some warm-up problems. The classical approach to estimate model parameters in various ML problems is to minimize a loss function using an optimization algorithm such as gradient descent (Ruder, 2016; Chen & Wainwright, 2015; Andrychowicz et al., 2016).Turning the focus to adversarially robust training, we work with the following optimization problem in supervised learning, which can be found in (Yin et al., 2019):

$$\hat{\mathbf{w}} = \arg\min_{\mathbf{w}} \frac{1}{n} \sum_{i=1}^{n} \sup_{\|\boldsymbol{\Delta}\| \leq \epsilon} l(\mathbf{x}^{(i)} + \boldsymbol{\Delta}, y^{(i)}, \mathbf{w}) \tag{4}$$

For unsupervised learning problems, one just removes the variables $y^{(i)}$ above. The optimization problem 4 depends on two variables: the adversarial perturbation $\boldsymbol{\Delta}$ and the model parameter $\mathbf{w}$. We solve for one variable assuming the other is given iteratively as illustrated in Algorithm 1. Specifically, we estimate $\boldsymbol{\Delta}^\star$ for robust learning by defining the worst case adversarial attack for a given parameter vector $\mathbf{w}^{(j-1)}$ ($j$ denotes the iteration number in gradient descent) and sample $\{\mathbf{x}^{(i)}, y^{(i)}\}$ as follows:

$$\boldsymbol{\Delta}^\star = \arg\sup_{\|\boldsymbol{\Delta}\| \leq \epsilon} l(\mathbf{x}^{(i)} + \boldsymbol{\Delta}, y^{(i)}, \mathbf{w}^{(j-1)})$$

For brevity, we drop the subscript $j - 1$ from the parameter $\mathbf{w}$ when it is clear from the context that the optimization problem is being solved for a particular iteration. Naturally, computing $\boldsymbol{\Delta}^\star$ by solving

---

**Input:** $\{\mathbf{x}^{(i)}, y^{(i)}\}$ for $i \in [n]$, $T$: number of iterations, $\eta_j$ : step size for iteration $j \in [T]$
$\mathbf{w}^{(0)} \leftarrow$ initial value ;
**for** $j = 1$ *to* $T$ **do**
$\quad$ gradient $\leftarrow 0$ ;
$\quad$ **for** $i = 1$ *to* $n$ **do**
$\quad\quad$ $\boldsymbol{\Delta}^\star = \arg\sup_{\|\boldsymbol{\Delta}\| \leq \epsilon} l(\mathbf{x}^{(i)} + \boldsymbol{\Delta}, y^{(i)}, \mathbf{w}^{(j-1)})$;
$\quad\quad$ gradient $\leftarrow$ gradient $+ \frac{\partial l(\mathbf{x}^{(i)} + \boldsymbol{\Delta}^\star, y^{(i)}, \mathbf{w}^{(j-1)})}{\partial \mathbf{w}}$;
$\quad$ **end**
$\quad$ $\mathbf{w}^{(j)} \leftarrow \mathbf{w}^{(j-1)} - \eta_j \frac{1}{n}$gradient
**end**
**Output:** $\hat{\mathbf{w}} = \mathbf{w}^{(T)}$

**Algorithm 1:** Plug and play algorithm

---

another maximization problem every time might not be necessarily efficient. To tackle this issue, we provide plug-and-play solutions of $\boldsymbol{\Delta}^\star$ for a given $(\mathbf{x}^{(i)}, y^{(i)}, \mathbf{w}^{(j)})$ where $i \in [n]$ and $j \in [T]$ for various widely-used ML problems.

## 3.1   Linear Regression

We start with a linear regression model which is used in various applications across numerous domains such as biology (Schneider et al., 2010), econometrics, epidemiology, and finance (Myers & Myers, 1990). As discussed in the previous sub-section, the adversary tries to perturb each sample to the maximum possible extent using the budget $\epsilon$ by solving the following maximization problem for each sample:

$$\boldsymbol{\Delta}^\star = \arg\sup_{\|\boldsymbol{\Delta}\| \leq \epsilon} \left( \mathbf{w}^\intercal \left( \mathbf{x}^{(i)} + \boldsymbol{\Delta} \right) - y^{(i)} \right)^2 \tag{5}$$

where $y^{(i)} \in \mathbb{R}$, $\mathbf{x}^{(i)}, \boldsymbol{\Delta} \in \mathbb{R}^d$ and $||\boldsymbol{\Delta}||$ denotes any general norm. We provide the following theorem to compute $\boldsymbol{\Delta}^\star$ in closed form.

**Theorem 3.** *For any general norm $\|\cdot\|$, the solution for problem in Eq.* (5) *for a given $\left(\mathbf{x}^{(i)}, y^{(i)}\right)$ is:*

$$\mathbf{\Delta}^{\star} = \begin{cases} \pm\epsilon\frac{\mathbf{v}}{\|\mathbf{v}\|}, & \mathbf{w}^{\mathsf{T}}\mathbf{x}^{(i)} - y^{(i)} = 0 \\ \operatorname{sign}(\mathbf{w}^{\mathsf{T}}\mathbf{x}^{(i)} - y^{(i)})\epsilon\frac{\mathbf{v}}{\|\mathbf{v}\|} & \mathbf{w}^{\mathsf{T}}\mathbf{x}^{(i)} - y^{(i)} \neq 0 \end{cases}$$

*where $\mathbf{v} \in \partial\|\mathbf{w}\|_{*}$ as specified in Definition 2.*

### 3.2 Logistic Regression

Next, we tackle logistic regression which is widely used for classification tasks in many fields such as medical diagnosis (Truett et al., 1967), marketing (Michael & Gordon, 1997) and biology (Freedman, 2009). Using previously introduced notations, we formulate logistic regression (Kleinbaum et al., 2002) with worst case adversarial attack in the following way:

$$\mathbf{\Delta}^{\star} = \underset{||\mathbf{\Delta}|| \leq \epsilon}{\arg\sup} \ \log\left(1 + \exp\left(-y^{(i)}\mathbf{w}^{\mathsf{T}}\left(\mathbf{x}^{(i)} + \mathbf{\Delta}\right)\right)\right) \tag{6}$$

where $y^{(i)} \in \{-1, 1\}$ and $\mathbf{x}^{(i)}, \mathbf{\Delta} \in \mathbb{R}^{d}$. The optimal solution for above optimization problem is provided in the following theorem.

**Theorem 4.** *For any general norm $\|\cdot\|$, and the problem specified in Eq.* (6)*, the optimal solution is given by $\mathbf{\Delta}^{\star} = -\epsilon y^{(i)}\mathbf{v}/\|\mathbf{v}\|$, where $\mathbf{v} \in \partial\|\mathbf{w}\|_{*}$ as specified in Definition 2.*

Theorem 21 presented in Section A.3 of appendix discusses a similar result for the hinge loss.

## 4 Main Results

### 4.1 Two-Layer Neural Networks

Consider a two-layer neural network for a binary classification problem with any general (convex or nonconvex) activation function $\sigma : \mathbb{R} \to \mathbb{R}$ in the first layer. As we work on the classification problem, we consider the log sigmoid activation function in the final layer. The general adversarial problem can be stated as:

$$\mathbf{\Delta}^{\star} = \underset{||\mathbf{\Delta}|| \leq \epsilon}{\arg\sup} \ \log\left(1 + \exp\left(-y^{(i)}\mathbf{v}^{\mathsf{T}}\sigma_{h}\sigma\left(\mathbf{W}^{\mathsf{T}}\left(\mathbf{x}^{(i)} + \mathbf{\Delta}\right)\right)\right)\right) \tag{7}$$

where $y^{(i)} \in \{1, -1\}$ for binary classification, $\mathbf{x}, \mathbf{\Delta} \in \mathbf{R}^{d}$, and the weight parameters $\mathbf{W} \in \mathbb{R}^{h \times d}$, and $\mathbf{v} \in \mathbb{R}^{h}$. Note that $h$ denotes the number of hidden units in the first layer, and the output for the general activation function $\sigma_{h}\sigma : \mathbb{R}^{h} \to \mathbb{R}^{h}$ is obtained by applying $\sigma : \mathbb{R} \to \mathbb{R}$ to each dimension independently. The optimal solution to the above problem is the following theorem.

**Theorem 5.** *For any general norm $\|\cdot\|$, and any activation function, the optimal solution for the problem specified in Eq.* (7) *can be solved using difference of convex functions.*

*Proof.* As $\log(\cdot)$ and $\exp(\cdot)$ are monotonically increasing functions, the adversarial problem specified in Eq. (7) can be equivalently expressed as:

$$\mathbf{\Delta}^{\star} = \underset{||\mathbf{\Delta}|| \leq \epsilon}{\arg\min} \ f(\mathbf{\Delta}) = y\mathbf{v}^{\mathsf{T}}\sigma_{h}\sigma\left(\mathbf{W}^{\mathsf{T}}\left(\mathbf{x} + \mathbf{\Delta}\right)\right) \tag{8}$$

where we have dropped the subscript (i) for brevity, as it is clear that the above problem is solved for each sample. The objective function of the above problem can be equivalently represented as:

$$f(\mathbf{\Delta}) = \sum_{i:y\mathbf{v}_i>0} y\mathbf{v}_i\sigma\left(\mathbf{z}_i\right) - \sum_{i:y\mathbf{v}_i<0} |y\mathbf{v}_i|\sigma\left(\mathbf{z}_i\right), \qquad \mathbf{z}_i = \mathbf{W}_i^{\mathsf{T}}\left(\mathbf{x} + \mathbf{\Delta}\right) \tag{9}$$

where $\mathbf{W}_i$ represents the $i^{\text{th}}$ row of matrix $\mathbf{W}$. Further, we express any general activation function $\sigma(\cdot)$ (which may be non-convex) as the difference of two convex functions using $\sigma(\mathbf{\Delta}) = \sigma_1(\mathbf{\Delta}) - \sigma_2(\mathbf{\Delta})$. Using

this formulation, the objective function in Eq. (9) can be expressed as $f(\boldsymbol{\Delta}) = g(\boldsymbol{\Delta}) - h(\boldsymbol{\Delta})$, where $g(\boldsymbol{\Delta})$ and $h(\boldsymbol{\Delta})$ are convex functions defined as:

$$g(\boldsymbol{\Delta}) = \sum_{i:y\mathbf{v}_i>0} y\mathbf{v}_i\sigma_1(\mathbf{z}_i) + \sum_{i:y\mathbf{v}_i<0} |y\mathbf{v}_i|\sigma_2(\mathbf{z}_i) \tag{10}$$

$$h(\boldsymbol{\Delta}) = \sum_{i:y\mathbf{v}_i>0} y\mathbf{v}_i\sigma_2(\mathbf{z}_i) + \sum_{i:y\mathbf{v}_i<0} |y\mathbf{v}_i|\sigma_1(\mathbf{z}_i) \tag{11}$$

where $\mathbf{z}_i$ is defined in Eq. (9). It should be noted that $g(\boldsymbol{\Delta})$ and $h(\boldsymbol{\Delta})$ are convex functions as they are positive weighted combination of convex functions $\sigma_1(\cdot)$ and $\sigma_2(\cdot)$. As the objective function $f(\boldsymbol{\Delta})$ can be expressed as difference of convex functions for any activation function specified in Appendix D, we can use difference of convex functions algorithms (DCA) (Tao & An, 1997). □

If set $\mathcal{S} = \{i \mid y\mathbf{v}_i < 0, i \in [h]\} = \emptyset$ and we have an activation function $\sigma(\boldsymbol{\Delta})$ such that $\sigma_2(\boldsymbol{\Delta}) = 0$, then $h(\boldsymbol{\Delta}) = 0$ and the problem in Eq. (7) reduces to a convex optimization problem. This may not be the case in general for two-layer neural networks. Therefore we use the difference of convex programming approach (Tao & An, 1997; Sriperumbudur & Lanckriet, 2009; Yu et al., 2021; Abbaszadehpeivasti et al., 2023; Le Thi et al., 2009; Yen et al., 2012; Khamaru & Wainwright, 2018; Nitanda & Suzuki, 2017) which are proved to converge to a critical point.

The first step to solve this optimization problem is constructing the functions $g(\boldsymbol{\Delta})$ and $h(\boldsymbol{\Delta})$, which requires decomposing the activation functions as the difference of convex functions. In order to do this, we decompose various activation functions commonly used in the literature as a difference of two convex functions. The decomposition is done by constructing a linear approximation of the activation function around the point where it changes the curvature. (~~refer~~Please see Appendix D).

Further, we compute $\boldsymbol{\Delta}^\star$ defined in Eq. (8) by expressing $f(\boldsymbol{\Delta}) = g(\boldsymbol{\Delta}) - h(\boldsymbol{\Delta})$ and using concave-convex procedure (Sriperumbudur & Lanckriet, 2009) or difference of convex function algorithm (DCA) (Tao & An, 1997). These algorithms are established to converge to a critical point, and hence the obtained $\boldsymbol{\Delta}^\star$ is plugged in Algorithm 1.

## 4.2 Learning Gaussian Graphical Models

Next, we provide a robust adversarial training process for learning Gaussian graphical models. These models are used to study the conditional independence of jointly Gaussian continuous random variables. This can be analyzed by inspecting the zero entries in the inverse covariance matrix, popularly referred as the precision matrix and denoted by $\boldsymbol{\Omega}$ (Lauritzen, 1996; Honorio et al., 2012). The classical (non-adversarial) approach (Yuan & Lin, 2007) solves the following optimization problem to estimate $\boldsymbol{\Omega}$ :

$$\boldsymbol{\Omega}^\star = \underset{\boldsymbol{\Omega}\succ 0}{\arg\min} - \log(\det(\boldsymbol{\Omega})) + \frac{1}{n}\sum_{i=1}^n \mathbf{x}^{(i)\intercal}\boldsymbol{\Omega}\mathbf{x}^{(i)} + c\,\|\boldsymbol{\Omega}\|_1$$

where $\boldsymbol{\Omega}$ is constrained to be a symmetric positive definite matrix and $c$ is a positive regularization constant. As the first term $\log(\det(\boldsymbol{\Omega}))$ in the above equation can not be influenced by adversarial perturbation in $\mathbf{x}^{(i)}$, we define the adversarial attack problem for this case as maximizing the second term by perturbing $\mathbf{x}^{(i)}$ for each sample:

$$\boldsymbol{\Delta}^\star = \underset{||\boldsymbol{\Delta}||\leq\epsilon}{\arg\sup} \left(\mathbf{x}^{(i)} + \boldsymbol{\Delta}\right)^\intercal \boldsymbol{\Omega} \left(\mathbf{x}^{(i)} + \boldsymbol{\Delta}\right) \tag{12}$$

For the above problem, we provide solutions for the $\ell_2$ and $\ell_\infty$ norm constraints as follows.

**Theorem 6.** *The solution for the problem in Eq. (12) with $\ell_2$ constraint on $\boldsymbol{\Delta}$ is given by $\boldsymbol{\Delta}^\star = (\mu^\star\mathbf{I} - \boldsymbol{\Omega})^{-1}\boldsymbol{\Omega}\mathbf{x}^{(i)}$, where $\mu^\star$ can be derived from the following ~~1 d~~1-D optimization problem:*

$$\max \quad -\frac{1}{2}\mathbf{x}^{(i)\intercal}\boldsymbol{\Omega}\left(\mu\mathbf{I} - \boldsymbol{\Omega}\right)^{-1}\boldsymbol{\Omega}\mathbf{x}^{(i)} - \frac{\mu\epsilon^2}{2}, \qquad such\ that \quad \mu\mathbf{I} - \boldsymbol{\Omega} \succeq 0 \tag{13}$$

**Theorem 7.** *The solution for the problem specified in Eq.* (12) *with $\ell_\infty$ constraint on $\boldsymbol{\Delta} \in \mathbb{R}^p$ can be derived from the last column/row of $\mathbf{Y}$ obtained from the following optimization problem:*

$$\max \left\langle \begin{bmatrix} \boldsymbol{\Omega} & \boldsymbol{\Omega}\mathbf{x}^{(i)} \\ (\boldsymbol{\Omega}\mathbf{x}^{(i)})^\mathsf{T} & 0 \end{bmatrix}, \mathbf{Y} \right\rangle \qquad such\ that\ \mathbf{Y}_{p+1,p+1} = 1,\ \mathbf{Y} \succeq 0,\ |\mathbf{Y}_{ij}| \leq \epsilon^2\ \forall i,j \in [p]$$

The results in Theorem 6 and Theorem 7 do not have a closed form but can be computed easily by solving a standard one-dimensional optimization problem and a SDP respectively. Very efficient scalable SDP solvers exist in practice (Yurtsever et al., 2021).

## 4.3 Matrix Completion

Assume we are given a partially observed matrix $\mathbf{X}$. Let $\mathcal{P}$ be a set of indices where the entries of $\mathbf{X}$ are observed (i.e., not missing). The classical (non-adversarial) matrix completion approach aims to find a low rank matrix (Shamir & Shalev-Shwartz, 2011) with small squared error in the observed entries:

$$\min_{\mathbf{Y}} \sum_{(i,j)\in\mathcal{P}} (\mathbf{X}_{ij} - \mathbf{Y}_{ij})^2 + c\|\mathbf{Y}\|_{\mathrm{tr}}$$

where $c$ is a positive regularization constant and $\|\cdot\|_{\mathrm{tr}}$ denotes trace norm of matrix which ensures low-rankness. Note that regularization does not impact the adversarial training framework. We define the following worst-case adversarial attack problem:

$$\boldsymbol{\Delta}^\star = \arg\sup_{||\boldsymbol{\Delta}||\leq\epsilon} \sum_{(i,j)\in\mathcal{P}} (\mathbf{X}_{ij} + \boldsymbol{\Delta}_{ij} - \mathbf{Y}_{ij})^2 \tag{14}$$

The solution for the above problem for the Frobenius norm constraint and entry-wise $\ell_\infty$ constraint on $\boldsymbol{\Delta}$ is proposed in Theorem 8 and Corollary 9.

**Theorem 8.** *The optimal solution for the optimization in Eq.* (14) *with Frobenius norm constraint on $\boldsymbol{\Delta}$ if $\exists (i,j) \in \mathcal{P}$ such that $\mathbf{X}_{ij} \neq \mathbf{Y}_{ij}$ is given by*

$$\boldsymbol{\Delta}_{ij}^\star = \begin{cases} \epsilon \dfrac{(\mathbf{X}_{ij}-\mathbf{Y}_{ij})}{\sqrt{\displaystyle\sum_{(i,j)\in\mathcal{P}} (\mathbf{X}_{ij}-\mathbf{Y}_{ij})^2}} & (i,j) \in \mathcal{P} \\[6pt] 0 & (i,j) \notin \mathcal{P} \end{cases}$$

*If $\mathbf{X}_{ij} = \mathbf{Y}_{ij}, \forall (i,j) \in \mathcal{P}$, then the optimal $\boldsymbol{\Delta}^\star$ can be any solution satisfying $\sum_{(i,j)\in\mathcal{P}} \boldsymbol{\Delta}_{ij}^2 = \epsilon$.*

**Corollary 9.** *The optimal solution for the optimization problem in Eq.* (14) *with the constraint $|\boldsymbol{\Delta}_{ij}| \leq \epsilon$ for all $(i,j) \in \mathcal{P}$ is given by $\boldsymbol{\Delta}_{ij} = \frac{(\mathbf{X}_{ij}-\mathbf{Y}_{ij})}{|\mathbf{X}_{ij}-\mathbf{Y}_{ij}|}\epsilon$.*

## 4.4 Max-Margin Matrix Completion

We start the discussion from the problem under the classical (non-adversarial) setting. Consider a partially observed label matrix where the observed entries are $+1$ or $-1$. Let $\mathcal{P}$ be the indices of the observed entries. The problem of max-margin matrix completion (Srebro et al., 2004) is defined as follows:

$$\min_{\mathbf{Y}} \sum_{(i,j)\in\mathcal{P}} \max(0, 1 - \mathbf{X}_{ij}\mathbf{Y}_{ij}) + c\,||\mathbf{Y}||_{\mathrm{tr}} \tag{15}$$

where $c > 0$ is a regularization constant and $||\cdot||_{\mathrm{tr}}$ represents the trace norm (Bach, 2008). As the second term, $||\mathbf{Y}||_{\mathrm{tr}}$ in the above optimization problem can not be affected by the adversary, we define the worst-case adversarial attack problem as the maximization of the first term with $\epsilon$ radius around $\mathbf{X}$:

$$\boldsymbol{\Delta}^\star = \arg\sup_{||\boldsymbol{\Delta}||\leq\epsilon} \sum_{(i,j)\in\mathcal{P}} \max(0, 1 - (\mathbf{X}_{ij} + \boldsymbol{\Delta}_{ij})\,\mathbf{Y}_{ij}) \tag{16}$$

The optimal $\boldsymbol{\Delta}^\star$ under Frobenius norm constraint on $\boldsymbol{\Delta}$ is proposed in the following theorem.

Table 2: A summary of our results for Rademacher complexity of various loss functions and norm constraints. $\epsilon$ is the radius of the norm constraint, $n$ is the number of training samples for regression and classification, $|\mathcal{P}|$ is the number of observed entries in matrix completion. The empirical Rademacher complexity is formally described in Definition 12 for regression and classification, and in Definition 15 for matrix completion. Lipschitz losses include the logistic loss, the hinge loss, the hyperbolic tangent, the logit, among others.

| | Problem | Loss function | Norm constraint | Prior results | Our result | Lower bound of $\hat{\mathfrak{R}}_S(\tilde{\mathcal{F}})$ | Upper bound of $\hat{\mathfrak{R}}_S(\tilde{\mathcal{F}})$ |
|---|---|---|---|---|---|---|---|
| Main results | Regression | Squared loss | Any norm | None | Theorem 13 | $\hat{\mathfrak{R}}_S(\mathcal{F}) - \mathcal{O}(\epsilon/n + \epsilon^2/\sqrt{n})$ | $\hat{\mathfrak{R}}_S(\mathcal{F}) + \mathcal{O}(\epsilon/n + \epsilon^2/\sqrt{n})$ |
| | Classification | Lipschitz loss | Any norm | $\ell_\infty$ norm (Yin et al., 2019) | Theorem 14 | $\max(\hat{\mathfrak{R}}_S(\mathcal{F}), \mathcal{O}(\epsilon/\sqrt{n}))$ | $\hat{\mathfrak{R}}_S(\mathcal{F}) + \mathcal{O}(\epsilon/\sqrt{n})$ |
| | | Lipschitz loss | Any norm | $\ell_p$ norm (Awasthi et al., 2020) | | | |
| | Matrix Completion | Squared loss | Entry-wise $\ell_\infty$ | None | Theorem 16 | $\hat{\mathfrak{R}}_S(\mathcal{F}) - \mathcal{O}(\epsilon/|\mathcal{P}|)$ | $\hat{\mathfrak{R}}_S(\mathcal{F}) + \mathcal{O}(\epsilon/|\mathcal{P}|)$ |
| | Max-Margin MC | Lipschitz loss | Entry-wise $\ell_\infty$ | None | Theorem 17 | $\hat{\mathfrak{R}}_S(\mathcal{F}) - \mathcal{O}(\epsilon/|\mathcal{P}|)$ | $\hat{\mathfrak{R}}_S(\mathcal{F}) + \mathcal{O}(\epsilon/|\mathcal{P}|)$ |
| | | Lipschitz loss | Entry-wise $\ell_\infty$ | None | Corollary 18 | $\max(\hat{\mathfrak{R}}_S(\mathcal{F}), \mathcal{O}(\epsilon/|\mathcal{P}|))$ | $\hat{\mathfrak{R}}_S(\mathcal{F}) + \mathcal{O}(\epsilon/|\mathcal{P}|)$ |

**Theorem 10.** *For the problem in Eq.* (16) *with Frobenius norm constraint on* $\mathbf{\Delta}$*, the solution is:*

$$\mathbf{\Delta}_{ij} = \begin{cases} -\mathbf{Y}_{ij} \dfrac{\epsilon}{\sqrt{\sum_{(i,j)\in\mathcal{P}} \mathbf{Y}_{ij}^2}} & (i,j) \in \mathcal{P}_1 \\[2em] 0 & (i,j) \notin \mathcal{P}_1 \end{cases}$$

*where* $\mathcal{P}_1 \subseteq \mathcal{P}$ *is chosen by sorting* $\mathbf{X}_{ij}\mathbf{Y}_{ij}$ *and selecting indices which satisfy* $\mathbf{X}_{ij}\mathbf{Y}_{ij} < 1 + \epsilon/\sqrt{\sum_{(i,j)\in\mathcal{P}} \mathbf{Y}_{ij}^2}$.

Similarly, the solution for the problem in Eq. (16) for the entry-wise $\ell_\infty$ norm is proposed as follows.

**Corollary 11.** *For the problem in Eq.* (16) *with the constraint* $|\mathbf{\Delta}_{ij}| \leq \epsilon$ *for all* $(i,j) \in \mathcal{P}$*, the optimal solution is given by* $\mathbf{\Delta}_{ij} = -sign(\mathbf{Y}_{ij})\epsilon$.

## 5 Implications

### 5.1 Novel Rademacher complexities

The motivation behind deriving the bounds for the adversarial Rademacher complexity is briefly discussed here. As shown in Theorem 8 in (Bartlett & Mendelson, 2002), the upper bound for generalization (the gap between empirical risk and population/expected risk) is $\hat{\mathfrak{R}}_S(\mathcal{F}) + \mathcal{O}(1/\sqrt{n})$. Hence an upper bound of order $\mathcal{O}(1/\sqrt{n})$ for the adversarial Rademacher complexity allows obtaining a $\mathcal{O}(1/\sqrt{n})$ generalization bound. Also, as shown in Proposition 4.12 in (Wainwright, 2019), the lower bound for generalization is $\frac{1}{2}\hat{\mathfrak{R}}_S(\mathcal{F}) - \Omega(1/\sqrt{n})$. Hence a lower bound of order $\Omega(1/\sqrt{n})$ for the adversarial Rademacher complexity allows obtaining a $\Omega(1/\sqrt{n})$ generalization bound.

Before diving into theorems and their proofs, we summarize our theoretical statistical/generalization contribution on Rademacher complexity bounds in Table 2, and make some important observations:

- The bounds for adversarial Rademacher complexity presented in Theorem 13 for linear regression (for any general norm), Theorem 16 for matrix completion, Theorem 17 and Corollary 18 for max-margin matrix completion are novel.
- The bounds presented in Theorem 14 for linear classifiers are for any general norm can be seen as a generalization of the $\ell_\infty$ norm (Yin et al., 2019) or any particular $p$-norm (Awasthi et al., 2020).

We utilize the closed-form solutions for various adversarial problems presented in Section 3 and Section 4 to derive new bounds for the adversarial Rademacher complexity. To start with, we define the empirical Rademacher complexity formally.

**Definition 12** (Rademacher complexity for regression and classification)**.** The empirical Rademacher complexity of the hypothesis class $\mathcal{F}$ with respect to a data set $S = \left\{ \left( \mathbf{x}^{(1)}, y^{(1)} \right), \ldots, \left( \mathbf{x}^{(n)}, y^{(n)} \right) \right\}$ is defined as:

$$\hat{\mathfrak{R}}_S(\mathcal{F}) = \frac{1}{n} \mathbb{E}_\sigma \left[ \sup_{h \in \mathcal{F}} \left( \sum_{i=1}^{n} \sigma_i h \left( \mathbf{x}^{(i)}, y^{(i)} \right) \right) \right]$$

We denote the adversarial function class with $\tilde{\mathcal{F}}$. Further, we utilize Theorem 3 to derive upper and lower bounds of the adversarial Rademacher complexity for linear regression.

**Theorem 13** (Regression with squared loss)**.** *Let the function class* $\mathcal{F} = \left\{ \left( \mathbf{x}^{(i)}, y^{(i)} \right) \mapsto \left( \mathbf{w}^\intercal \mathbf{x}^{(i)} - y^{(i)} \right)^2 \mid \|\mathbf{w}\|_* \leq C \right\}$ *and the adversarial function class* $\tilde{\mathcal{F}} = \left\{ \left( \mathbf{x}^{(i)}, y^{(i)} \right) \mapsto \sup_{\|\mathbf{\Delta}\| \leq \epsilon} \left( \mathbf{w}^\intercal (\mathbf{x}^{(i)} + \mathbf{\Delta}) - y^{(i)} \right)^2 \mid \|\mathbf{w}\|_* \leq C \right\}$, *then:*

$$\hat{\mathfrak{R}}_S(\mathcal{F}) - b(n, \epsilon, C) \leq \hat{\mathfrak{R}}_S(\tilde{\mathcal{F}}) \leq \hat{\mathfrak{R}}_S(\mathcal{F}) + b(n, \epsilon, C), \quad where \; b(n, \epsilon, C) = \frac{2\epsilon C}{n} \mathbb{E}_\sigma \left[ \left\| \sum_{i=1}^{n} \sigma_i \mathbf{x}^{(i)} \right\| \right] + \frac{\epsilon^2 C^2}{\sqrt{n}}$$

It should be noted that the above theorem is proposed for any general norm and any further simplification of the term $\mathbb{E}_\sigma \left[ \left\| \frac{1}{n} \sum_{i=1}^{n} \sigma_i \mathbf{x}^{(i)} \right\| \right]$ requires a further assumption on the specific norm. For example, it can be easily bounded as $\frac{c_1}{\sqrt{n}}$ using Khintchine's inequality for the Euclidean norm, where $c_1$ is some constant.

*Proof sketch.* To prove the above theorem, we have used carefully manipulated the Rademacher complexity of adversarial function class in terms of classical function class with the help of Theorem 3. Further, we have used the Ledoux-Talagrand contraction principle for Lipchitz functions and Khintchine's inequality. A similar proof recipe is used to prove other theorems in this section, whose detailed proofs can be seen in Section C of supplementary material. □

Further, we utilize Theorem 4 or Theorem 21 to derive the upper and lower bounds of the adversarial Rademacher complexity for linear classification. Similar to the existing works (Yin et al., 2019; Awasthi et al., 2020) mentioned in Table 2 as well as prior work on non-adversarial regimes (Kakade et al., 2008), we analyze the Rademacher complexity of linear functions, which allows for bounding the Rademacher complexity of relatively more complex Lipschitz functions (e.g., logistic loss, hinge loss, hyperbolic tangent, logit) by using the Ledoux-Talagrand contraction lemma (Ledoux & Talagrand, 2013).

**Theorem 14** (Classification)**.** *Let the function class* $\mathcal{F} = \left\{ \left( \mathbf{x}^{(i)}, y^{(i)} \right) \mapsto -y^{(i)} \mathbf{w}^\intercal \mathbf{x}^{(i)} \mid \|\mathbf{w}\|_* \leq C \right\}$ *and the adversarial function class* $\tilde{\mathcal{F}} = \left\{ \left( \mathbf{x}^{(i)}, y^{(i)} \right) \mapsto \sup_{\|\mathbf{\Delta}\| \leq \epsilon} -y^{(i)} \mathbf{w}^\intercal (\mathbf{x}^{(i)} + \mathbf{\Delta}) \mid \|\mathbf{w}\|_* \leq C \right\}$, *then:*

$$\max \left\{ \hat{\mathfrak{R}}_S(\mathcal{F}), \frac{C\epsilon}{2\sqrt{2n}} \right\} \leq \hat{\mathfrak{R}}_S(\tilde{\mathcal{F}}) \leq \hat{\mathfrak{R}}_S(\mathcal{F}) + \frac{C\epsilon}{\sqrt{n}}$$

Similar results have been proposed in the literature for the $\ell_\infty$ norm (Yin et al., 2019) or any particular $p$-norm (Awasthi et al., 2020), whereas our result is more general, pertaining to any norm.

We further move on to the matrix completion problem and propose the following definition of the Rademacher complexity motivated from Definition 12.

**Definition 15** (Rademacher complexity for matrix completion)**.** The empirical Rademacher complexity of the hypothesis class $\mathcal{F}$ with respect matrix $\mathbf{X}$ with observed entries $(i, j) \in \mathcal{P}$ is defined as:

$$\hat{\mathfrak{R}}_\mathbf{X}(\mathcal{F}) = \frac{1}{n} \mathbb{E}_\sigma \left[ \sup_{h \in \mathcal{F}} \left( \sum_{(i,j) \in \mathcal{P}} \sigma_{ij} h(\mathbf{X}_{ij}) \right) \right]$$

Table 3: Error metrics on real-world data sets for various supervised and unsupervised ML problems. Notice that the proposed approach ~~outperforms the baselines~~performs better (most of the time) as compared to other methods ("No error", "Random", "FGSM", "PGD", and "TRADES"). MC and NN denote matrix completion and neural networks respectively.

| | Problem | Loss function | Dataset | Metric | Norm | No error | Random | FGSM | PGD | TRADES | Proposed |
|---|---|---|---|---|---|---|---|---|---|---|---|
| **Warm up** | Regression | Squared loss | BlogFeedback | MSE | Euclidean | 11.66 | 11.66 | 11.66 | 11.66 | 11.49 | 11.18 |
| | Regression | Squared loss | BlogFeedback | MSE | $\ell_\infty$ | 11.66 | 11.66 | 11.66 | 11.66 | 11.47 | 11.20 |
| | Classification | Logistic loss | ImageNet | Accuracy | Euclidean | 49.80 | 48.13 | 49.8 | 49.8 | 50.9 | 56.75 |
| | Classification | Logistic loss | ImageNet | Accuracy | $\ell_\infty$ | 49.80 | 45.46 | 49.8 | 49.8 | 52 | 55.34 |
| | Classification | Hinge loss | ImageNet | Accuracy | Euclidean | 47.89 | 46.66 | 47.8 | 47.9 | 52 | 52.31 |
| | Classification | Hinge loss | ImageNet | Accuracy | $\ell_\infty$ | 47.89 | 49.59 | 49.8 | 49.8 | 52 | 52 |
| **Main results** | Classification | NN: ~~ReLu~~ReLU | ImageNet | Accuracy | Euclidean | 70.74 | 70.66 | 53.88 | 54.48 | 71.55 | 76.49 |
| | Classification | NN: Sigmoid | ImageNet | Accuracy | Euclidean | 72.5 | 71.08 | 63.45 | 75.89 | 67.92 | 73.04 |
| | Graphical Model | Log-likelihood | TCGA | Likelihood | Euclidean | -7984.8 | -7980.6 | -7596.5 | -7596.5 | -7603.4 | -7406.1 |
| | Graphical Model | Log-likelihood | TCGA | Likelihood | $\ell_\infty$ | -7984.8 | -7984.4 | -7888.4 | -7916.2 | -7917.2 | -7918.7 |
| | Matrix Completion | Squared loss | Netflix | MSE | Frobenius | 4.78 | 4.89 | 4.6 | 3.9 | 4.2 | 3.2 |
| | Matrix Completion | Squared loss | Netflix | MSE | Entry-wise $\ell_\infty$ | 4.78 | 4.78 | 4.57 | 3.87 | 4.26 | 3.86 |
| | Max-Margin MC | Squared loss | HouseRep | Accuracy | Frobenius | 97.05 | 89.72 | 96.27 | 97.02 | 63.22 | 97.14 |
| | Max-Margin MC | Squared loss | HouseRep | Accuracy | Entry-wise $\ell_\infty$ | 92.4 | 60.7 | 73.28 | 75.46 | 39.99 | 92.5 |

We further utilize Corollary 9 to derive novel upper and lower bounds for the adversarial Rademacher complexity for matrix completion.

**Theorem 16** (Matrix completion with squared loss)**.** *Let the function class* $\mathcal{F} = \{\mathbf{X}_{ij} \mapsto (\mathbf{X}_{ij} - \mathbf{Y}_{ij})^2 \mid \|\mathbf{Y}\|_* \leq C\}$ *and the adversarial function class* $\tilde{\mathcal{F}} = \{\mathbf{X}_{ij} \mapsto \sup_{\|\mathbf{\Delta}_{ij}\| \leq \epsilon} (\mathbf{X}_{ij} + \mathbf{\Delta}_{ij} - \mathbf{Y}_{ij})^2 \mid \|\mathbf{Y}\|_* \leq C\}$, *then:*

$$\hat{\mathfrak{R}}_{\mathbf{X}}(\mathcal{F}) - \frac{2\epsilon C}{|\mathcal{P}|} \mathbb{E} \|\boldsymbol{\sigma}\| \leq \hat{\mathfrak{R}}_{\mathbf{X}}(\tilde{\mathcal{F}}) \leq \hat{\mathfrak{R}}_{\mathbf{X}}(\mathcal{F}) + \frac{2\epsilon C}{|\mathcal{P}|} \mathbb{E} \|\boldsymbol{\sigma}\|$$

*where* $\boldsymbol{\sigma}$ *is a matrix whose entries* $\boldsymbol{\sigma}_{ij}$ *for* $(i,j) \in \mathcal{P}$ *follows Rademacher distribution.*

The term $\mathbb{E} \|\boldsymbol{\sigma}\|$ can be simplified for particular norms. For example, if we consider $\|\mathbf{Y}\|_1 \leq C$, then $\mathbb{E}_\sigma [\|\boldsymbol{\sigma}\|_\infty] = 1$ where $\|.\|_1$ and $\|.\|_\infty$ denote the entrywise $\ell_1$ and $\ell_\infty$ norm respectively.

Further, we utilize Corollary 11 to derive upper and lower bounds for the adversarial Rademacher complexity for max-margin matrix completion. As previously discussed before Theorem 14, prior work in classification on adversarial regimes (Yin et al., 2019; Awasthi et al., 2020) mentioned in Table 2 as well as prior work on non-adversarial regimes (Kakade et al., 2008) use the Rademacher complexity of linear functions to bound the Rademacher complexity of relatively more complex Lipschitz functions by using the Ledoux-Talagrand contraction lemma (Ledoux & Talagrand, 2013). We use the same principle for max-margin matrix completion which typically uses the hinge loss, which is Lipschitz.

**Theorem 17** (Max-margin matrix completion)**.** *Let the function class* $\mathcal{F} = \{\mathbf{X}_{ij} \mapsto -\mathbf{X}_{ij}\mathbf{Y}_{ij} \mid \|\mathbf{Y}\|_* \leq C\}$ *and the adversarial function class* $\tilde{\mathcal{F}} = \{\mathbf{X}_{ij} \mapsto \sup_{\|\mathbf{\Delta}_{ij}\| \leq \epsilon} - (\mathbf{X}_{ij} + \mathbf{\Delta}_{ij}) \mathbf{Y}_{ij} \mid \|\mathbf{Y}\|_* \leq C\}$, *then:*

$$\hat{\mathfrak{R}}_{\mathbf{X}}(\mathcal{F}) - \frac{\epsilon C}{|\mathcal{P}|} \mathbb{E} \|\boldsymbol{\sigma}\| \leq \hat{\mathfrak{R}}_{\mathbf{X}}(\tilde{\mathcal{F}}) \leq \hat{\mathfrak{R}}_{\mathbf{X}}(\mathcal{F}) + \frac{\epsilon C}{|\mathcal{P}|} \mathbb{E} \|\boldsymbol{\sigma}\|$$

*where* $\boldsymbol{\sigma}$ *is a matrix whose entries* $\boldsymbol{\sigma}_{ij}$ *for* $(i,j) \in \mathcal{P}$ *follows Rademacher distribution.*

**Corollary 18** (Max-margin matrix completion)**.** *For the function class defined in Theorem 17 with* $\ell_1$ *norm constraint, i.e.,* $\|\mathbf{Y}\|_1 \leq C$, *we obtain the following tighter lower bound:* $\max \left\{ \hat{\mathfrak{R}}_{\mathbf{X}}(\mathcal{F}), \epsilon C / |\mathcal{P}| \right\} \leq \hat{\mathfrak{R}}_{\mathbf{X}}(\tilde{\mathcal{F}})$.

## 5.2  Real-World Experiments

~~We~~As a sanity check, we compare the proposed approach on real-world datasets against five ~~baselines~~methods of having no adversary, a random adversary (Gilmer et al., 2019; Qin et al., 2021), and other well-known ~~baselines~~methods such as FGSM (Goodfellow et al., 2015), projected gradient descent (PGD), and TRADES (Zhang et al., 2019). While we could have used $\epsilon$-perturbed test data (coming from the same distribution of

the training data) with some synthetic adversary, we preferred to use a more challenging scenario: test data coming from a different distribution than the training data.

The results are summarized in Table 3, and it can be clearly observed that the proposed method ~~outperforms the baselines~~performs better (most of the time) as compared to other methods. Table 5 shows run time of the proposed method is comparable to ~~baselines (refer Appendix E).~~the one of other methods (most of the time). Please see Appendix E.

## 6  Concluding Remarks

We proposed a robust adversarial training framework which can be integrated with widely used ML models without any significant computational overhead. As adversarial attacks are not limited only to the problems covered in this work, our analysis can be extended to other problems such as clustering, discrete optimization problems, and randomized algorithms in the future.

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
