# OpenReview forum: "On Exact Solutions of the Inner Optimization Problem of Adversarial Robustness"
_TMLR — Rejected by TMLR_

### Review · Reviewer_FtWW · 2024-03-22

**Summary Of Contributions:**

The paper presents exact solutions to the problem of finding a optimal norm-bounded adversaries for several machine learning machine learning problems, including binary classification via the logistic and hinge losses, and some matrix completion problems.
Depending on the problem, these solutions can be obtained as closed-forms, or by solving auxiliary problems via SDP, difference of convex functions, or 1D optimization.
These solutions are used to prove novel results on the adversarial Rademacher complexity associated to these problems.
The paper includes empirical results relying on the derived adversaries.

**Audience:**

Yes

**Broader Impact Concerns:**

None.

**Claims And Evidence:**

No

**Requested Changes:**

If the authors wish to keep the claims pertaining to the empirical utility of the provided exact solutions, the experimental section needs substantial changes. In particular, relevant and unaltered benchmarks from the literature need to be employed, without any distribution shift at testing time. The metrics should be: standard performance on the training set, and performance under strong adversarial attacks (both the derived exact solutions and iterative schemes such as PGD) within the considered threat models.

If this is not possible, I think removing the experimental section completely, along with the associated claims would be fine. The paper would then become exclusively theoretical and presenting the novel generalization bounds.

**Strengths And Weaknesses:**

**Strenghts.**

The Rademacher complexity results, obtained exploiting some of the derived exact solutions, appear to be novel and of potential interest to part of the adversarial ML community.

**Weaknesses.**

Given that the authors claim to show (in the abstract) "significant performance improvement on real-world datasets over various gradient ascent based baselines", I find the provided empirical results to be extremely sketchy and misleading. This overshadows the theoretical contributions.

For instance, the authors state that they provide ImageNet results without further clarifications, whereas the appendix makes it clear that they selected two ImageNet classes to perform binary classification and only trained on the relevant samples. The test set is instead chosen to be about two different classes, unseen at training time. This is not a standard benchmark or evaluation scheme by any means, and is arguably quite unrelated to the proposed results or adversaries (which operate on local norm-bounded perturbations). It is hence unsurprising that many models behave similarly to tossing a fair coin (50% accuracy).  As far as I understand, no adversary is deployed at evaluation time either, adding to the disconnect.
Furthermore, given that some of the exact solutions require iterative schemes, such as difference-of-convex programming for the two-layer networks, it is unclear how it is possible that this would be more efficient than cheap adversaries such as FGSM or few PGD steps.

Given the above points on the experiments, the utility of some of the provided exact solutions (for instance, for two-layer NNs) is unclear, as some of them are not employed to compute the novel Rademacher complexities either. It is also very misleading to treat the cost of solving, for instance, an SDP problem, as constant (as claimed in the introduction, when stating complexities).

Finally, it is unclear if some of the adversarial problems (for instance, those on matrix completion) are defined here for the first time, or if they come from the literature. In both cases, more motivation is required (for instance, showing that standard models are vulnerable to adversaries in the employed threat models).

---

> ### Author Response · Authors · 2024-05-08
>
> > The Rademacher complexity results... appear to be novel and of potential interest to part of the adversarial ML community.
>
> We thank the reviewer for the appreciation of our work.
>
> > ... "significant performance improvement on real-world datasets over various gradient ascent based baselines"... This overshadows the theoretical contributions.
>
> We have toned down our claims regarding the experiments in our manuscript. In the abstract, end of section 1, and beginning of Section 5.2, we have mentioned that our experiments are a "sanity check". We have also removed words such as "outperform" or "baseline" throughout the manuscript to avoid misunderstandings of our goal.
>
> > ... the appendix makes it clear that they selected two ImageNet classes to perform binary classification and only trained on the relevant samples. The test set is instead chosen to be about two different classes, unseen at training time...
>
> As mentioned in Section 5.2: "While we could have used $\epsilon$-perturbed test data (coming from the same distribution of the training data) with some synthetic adversary, we preferred to use a more challenging scenario: test data coming from a different
> distribution than the training data."
>
> > ... misleading to treat the cost of solving, for instance, an SDP problem, as constant...
>
> We have removed paragraphs 4 and 5 which started from an older version, since they put too much emphasis to the computational/optimization side, which we already argued in the second paragraph. Having said that, please see Table 5 in which runtimes are better than other methods (most of the time).
>
> > ... unclear if some of the adversarial problems (for instance, those on matrix completion) are defined here for the first time... more motivation is required...
>
> As we mentioned in Page 2: "We use the adversarially robust training framework of Yin et al. (2019)". The idea behind eq.(4) can be found in Yin et al. (2019). Strictly speaking, eq.(4) corresponds to supervised learning problems (having $x^{(i)}$ and $y^{(i)}$), which is more general than what applies to unsupervised problems (having only $x^{(i)}$). We briefly clarified this in our manuscript around eq.(4).
>
> In our manuscript, supervised problems include classification and regression, and unsupervised problems include Gaussian graphical modes, matrix factorization, max-margin matrix factorization.
>
> > If the authors wish to keep... the experimental section... without any distribution shift at testing time...
> > If this is not possible... removing the experimental section completely... would be fine.
>
> We hope we have addressed the reviewers' concerns but we are open to more feedback to improve our work.
>
> Finally, we noticed that the reviewer chose "Claims And Evidence: No". Please let us know if there is anything else that we are missing.

---

### Review · Reviewer_dGaG · 2024-04-17

**Summary Of Contributions:**

The paper considers the problem of creating adversarial examples for various settings. It begins by examining linear models in both classification and regression settings, where it exactly solves the adversarial examples generation problem. It then proceeds to two-layer neural networks, formulating the problem as a difference of convex functions problem. Furthermore, it explores both graphical models and matrix completion settings. The authors utilize their results to integrate the derived solution into the standard adversarial training loop. Finally, they derive generalization bounds for the adversarial risk in both linear and matrix completion settings based on the proposed solutions.

**Audience:**

Yes

**Broader Impact Concerns:**

The authors did not include a Broader Impact statement in their manuscript. While the proposed methods aim to generate adversarial examples, potentially rendering them susceptible to misuse, the authors propose methods to harden models against such attacks.

**Claims And Evidence:**

No

**Requested Changes:**

Please consider the weakness section.

**Strengths And Weaknesses:**

Strengths:
- The paper is well-written.
- The problem addressed is of great importance and interest.
- The results regarding matrix completion appear to be novel to the best of my knowledge.

Weaknesses:
- The authors' claims are sometimes overstated. For example, in the fourth paragraph of the introduction, the claim about reducing the runtime from $O(nTL)$ to $O(nT)$ holds true only for linear models. Similarly, in the fifth paragraph, the assertion about the approach being practical for problems with complex objective functions does not hold true for neural networks.
- The paper lacks thorough comparison to existing literature. For instance, the closed-form solution for $L_P$ attacks in linear models was already addressed in [1], and the use of difference of convex solutions for neural network attacks was previously explored in [2].
- The experimental settings lack sufficient detail. How was the evaluation performed? Are the reported numbers indicative of empirical adversarial risk? If so, which algorithm was used for generating testset adversarial examples?

[1] Awasthi, Pranjal, Natalie Frank, and Mehryar Mohri. "Adversarial learning guarantees for linear hypotheses and neural networks." International Conference on Machine Learning. PMLR, 2020.
[2] Awasthi, P., Mao, A., Mohri, M., & Zhong, Y. (2024). DC-programming for neural network optimizations. Journal of Global Optimization, 1-17.

---

> ### Author Response · Authors · 2024-05-08
>
> > - The paper is well-written.
> > - The problem addressed is of great importance and interest.
> > - The results regarding matrix completion appear to be novel to the best of my knowledge.
>
> We thank the reviewer for the appreciation of our work.
>
> > - ... in the fourth paragraph of the introduction, the claim about reducing the runtime from $O(nTL)$ to $O(nT)$ holds true only for linear models...
> > ... in the fifth paragraph, the assertion about the approach being practical for problems with complex objective functions does not hold true for neural networks.
>
> We have removed paragraphs 4 and 5 which started from an older version, since they put too much emphasis to the computational/optimization side, which we already argued in the second paragraph. Having said that, please see Table 5 in which runtimes are better than other methods (most of the time).
>
> > - The paper lacks thorough comparison to existing literature... the closed-form solution for $L_P$ attacks in linear models was already addressed in [1]...
>
> We have already properly referred (Awasthi et al. 2020). Please see and the top of Page 9. Prior results for classification with Lipschitz losses focused exclusively on $\ell_p$ norm constraints ($\ell_\infty$ in Yin et al. 2019 and $\ell_p$ in Awasthi et al. 2020). Our result in Theorem 14 is not only for $\ell_p$ norms, but for any norm, i.e., any real-valued function satisfying triangle inequality, absolute homogeneity and positiveness.
>
> This includes $\ell_p$ norms such as the $\ell_1$ norm but also norms such as the k-support norm (Argyriou et al. Sparse prediction with the k-support norm. NeurIPS, 2012), the $\ell_{1,2}$ overlapping norm (Jacob et al. Group lasso with overlap and graph lasso. NeurIPS, 2009) and the $\ell_{1,\infty}$ overlapping norm (Mairal et al. Network flow algorithms for structured sparsity. NeurIPS, 2010) to mention a few. If wanted, we can include this in the manuscript.
>
> > ... the use of difference of convex solutions for neural network attacks was previously explored in [2].
>
> Our second submission of our work which included neural networks was made to AISTATS 2023 with deadline on October 2022. We believe Awasthi and coauthors were concurrently working on  similar ideas. The main difference is that they only provided a result or the ReLU activation (see lines after eq.(3) in Awasthi et al. 2024). ReLU is convex and thus, the result in Theorem 1 in (Awasthi et al. 2024) relies on this. Our result is for any activation function which may be convex or nonconvex (please see Appendix D and Table 4). Finally, our result also accounts for the $\log(1+\exp(\cdot))$ composition. We have updated Table 1.
>
> > - The experimental settings lack sufficient detail...
>
> As mentioned in Section 5.2: "While we could have used $\epsilon$-perturbed test data (coming from the same distribution of the training data) with some synthetic adversary, we preferred to use a more challenging scenario: test data coming from a different
> distribution than the training data."
>
> Table 3 includes a "Metric" column, and Appendix E contains all the experimental details:
> - Regression: "We use mean square error (MSE) to evaluate the performance on a test set".
> - Classification: "We use the accuracy metric to evaluate the performance on a test set".
> - Gaussian Graphical models: "We compare the training approaches based on the log-likelihood of a test set from the learned precision matrices".
> - Matrix Completion: "We use the MSE metric on the test set".
> - Max-Margin Matrix Completion: "We use the percentage of correctly recovered votes on a test set".
>
> Finally, we noticed that the reviewer chose "Claims And Evidence: No". Please let us know if there is anything else that we are missing.

---

### Review · Reviewer_odqc · 2024-04-17

**Summary Of Contributions:**

This paper studies adversarial training (AT). The authors provide several closed form solutions/new forms for obtaining the pertubation in each round. They also apply the closed forms to get the relationship between the Radmarher complexity of the adversarial training problem and the original problem.

**Audience:**

Yes

**Claims And Evidence:**

Yes

**Requested Changes:**

I respect the authors' efforts, but I have major concerns about Sections 3 and 4. To be more specific, most of the closed forms are obtained by simply optimizing standard loss functions such as hinge loss, square loss, and logistic loss, and I do not think these can be claimed as contributions of any paper.

**Strengths And Weaknesses:**

I have major concerns on the contributions of this paper: In the first part (Sections 3-4), the authors provide several closed form solutions\new forms for computing the perturbation in AT. However, after a careful reading, and with respect to the authors, I have to say that, it seems to me that these results are trivial. To be more specific:

1. For the three results in Section 3, the closed form solutions are obtained by solving standard optimization problems. Note that, **these are standard loss functions**, like hinge loss, square loss, and logistic loss. How to optimize these functions are well-known, not limited to the AT community, but also to the general ML people, and it is surprising that the authors listed these as their contributions.

2. For Section 4, in 4.1 (two-layer NN), the authors merely changed the objective function in a different form, and I don't see the necessity of doing so; For Section 4.2, the problem is solving a **standard quadratic optimization** problem, which is also a trivial to solve; For the closed from in 4.3 and 4.4, again, the results are obtained by simply optimizing quadratic/hinge loss (in matrix form), which are standard to the ML community.

3. In Section 5, the results are relatively more interesting, but they are basically obtained by plugging in the closed form solution of the perturbation in the definition of Radmarher complexity (based on sup(A+B)\leq supA + supB, the extra term introduced by perturbation can basically be dealt with independently).

---

> ### Author Response · Authors · 2024-05-08
>
> > ... However, after a careful reading, and with respect to the authors, I have to say that, it seems to me that these results are trivial.
>
> We appreciate that the reviewer has expressed in an honest yet respectful fashion. We believe this encourages academic discussion. We somehow disagree with the reviewers' opinion and will try to argue in what follows.
>
> > 1. For the three results in Section 3, the closed form solutions are obtained by solving standard optimization problems... How to optimize these functions are well-known... it is surprising that the authors listed these as their contributions.
>
> Please note that Section 3 is called "Warm Up". We also make this distinction in Table 1. Having said that, prior results of (Xu et al., 2008) and (Liu et al., 2020) are only for the Euclidean norm constraints, which is a particular $\ell_p$ norm where $p=2$. Our result in Theorems 3, 4 and 21 is not only for $\ell_p$ norms, but for any norm, i.e., any real-valued function satisfying triangle inequality, absolute homogeneity and positiveness.
>
> This includes $\ell_p$ norms such as the $\ell_1$ norm but also norms such as the k-support norm (Argyriou et al. Sparse prediction with the k-support norm. NeurIPS, 2012), the $\ell_{1,2}$ overlapping norm (Jacob et al. Group lasso with overlap and graph lasso. NeurIPS, 2009) and the $\ell_{1,\infty}$ overlapping norm (Mairal et al. Network flow algorithms for structured sparsity. NeurIPS, 2010) to mention a few. If wanted, we can include this in the manuscript.
>
> > 2. For Section 4, in 4.1 (two-layer NN), the authors merely changed the objective function in a different form, and I don't see the necessity of doing so; ...
>
> The proof of Theorem 5 shows that eq.(7) and eq.(8) are equivalent (since $\log$ and $\exp$ are monotonically increasing functions). The objective function in eq.(8) is a difference of convex functions. This is why the change from eq.(7) to eq.(8) is useful.
>
> > 2. ... For the closed from in 4.3 and 4.4, again, the results are obtained by simply optimizing quadratic/hinge loss (in matrix form), which are standard to the ML community.
>
> We are not aware of any prior result related to matrix completion and max-margin matrix completion (see Table 1). If the reviewer knows of any reference, please let us know. Our results are useful to compute novel Rademacher complexities (see Table 4) or lead to a less intuitive sorting-based algorithm.
>
> > 3. In Section 5, the results are relatively more interesting, but they are basically obtained by plugging in the closed form solution...
>
> Table 2 shows that most of our results are novel (Theorems 13, 16, 17, Corollary 18), which we also discuss at the top of Page 9. Prior results for classification with Lipschitz losses focused exclusively on $\ell_p$ norm constraints ($\ell_\infty$ in Yin et al. 2019 and $\ell_p$ in Awasthi et al. 2020). Our result in Theorem 14 is not only for $\ell_p$ norms, but for any norm.
>
> > ... I have major concerns about Sections 3 and 4... I do not think these can be claimed as contributions of any paper.
>
> As argued before Sections 3 and 4 should be seen not by themselves but by their implications. In addition, we believe some machine learning papers should be appreciated not by the complexity of the mathematics (in the eyes of the beholder) but by their implications. Take for instance:
> - Papers which proposed using well-known methods (e.g., gradient descent and dropout regularization) for deep neural networks. See for instance (Krizhevsky et al. ImageNet Classification with Deep Convolutional Neural Networks, NeurIPS 2012).
> - Papers distilling well-known ways to obtain convergence rates. See for instance (Duchi et al. Efficient Learning using Forward-Backward Splitting, NeurIPS 2009) where the authors themselves state "Our algorithm is a distillation of known approaches for convex programming".
>
> In the examples above, it was not the complexity of the mathematics what made the papers being accepted, but (some initial signal of) their implications. Arguably, when those papers were accepted, it was very hard to know whether they would become very useful or not. Now, after many years, we know their impact.

---

### Decision · Action_Editor_PgvH · 2024-05-17

**Recommendation:** Reject

**Comment:**

The paper presents both empirical and theoretical analysis for adversarial training. The Rademacher analysis is very clean, and applies to a general norm. The paper is clearly written and easy to follow. However, the reviewers have some concerns regarding the contribution of the paper and these concerns are not well addressed in the discussion period.

Some reviewers mentioned that the empirical analysis is misleading as some nonstandard benchmark datasets are used in the experiments.  Also the details in the empirical analysis are somewhat missing.

Reviewers also mentioned that the paper considers simple loss functions such as least squares loss and logistic loss, which explains why they have closed-form solutions for the inner maximization problem. This closed-form solution makes it convenient to study the Rademacher complexity for adversarial training.

Reviewers also mentioned that the comparison with existing studies is lacking. For example, the difference of convex solutions for neural network attacks  has already been used in the literature.

Based on these comments, I have to not recommend the publication at this moment. I hope these comments would be useful for the authors to further improve the paper.

Some minor issues:
- In the proof of Theorem 13: $R_S(\tilde{F})=R_S(F)+$ should be $R_S(\tilde{F})\leq R_S(F)+$
- According to the proof of Theorem 13: it seems that $2\epsilon C$ should be $2\epsilon C^2$ in the last second displayed equation of the proof.

**Audience:**

Yes. Adversarial training is an important topic. The paper presents both theoretical and empirical analysis.

**Claims And Evidence:**

The theoretical analysis is correct and clearly stated. As some reviewers mentioned, the empirical analysis is misleading as the benchmark dataset is not standard.

**Resubmission Of Major Revision:**

The authors may consider submitting a major revision at a later time.